# Mechanical Strength and Conductivity of Cementitious Composites with Multiwalled Carbon Nanotubes: To Functionalize or Not?

**DOI:** 10.3390/nano14010080

**Published:** 2023-12-27

**Authors:** Edgar A. O’Rear, Suthisa Onthong, Thirawudh Pongprayoon

**Affiliations:** 1School of Sustainable Chemical, Biological and Materials Engineering, University of Oklahoma, Norman, OK 73019, USA; suthisa.on@gmail.com; 2Institute for Applied Surfactant Research, University of Oklahoma, Norman, OK 73019, USA; 3Department of Chemical Engineering, Faculty of Engineering, King Mongkut’s University of Technology North Bangkok, Bangkok 10800, Thailand; thirawudh.p@eng.kmutnb.ac.th; 4Center of Eco-Materials and Cleaner Technology, King Mongkut’s University of Technology North Bangkok, Bangkok 10800, Thailand

**Keywords:** cement, composites, carbon nanotubes, functionalization, concrete, compressive strength, mortar

## Abstract

The incorporation of carbon nanotubes into cementitious composites increases their compressive and flexural strength, as well as their electrical and thermal conductivity. Multiwalled carbon nanotubes (MWCNTs) covalently functionalized with hydroxyl and carboxyl moieties are thought to offer superior performance over bare nanotubes, based on the chemistry of cement binder and nanotubes. Anionic carboxylate can bind to cationic calcium in the hydration products, while hydroxyl groups participate in hydrogen bonding to anionic and nonionic oxygen atoms. Results in the literature for mechanical properties vary widely for both bare and modified filler, so any added benefits with functionalization are not clearly evident. This mini-review seeks to resolve the issue using an analysis of reports where direct comparisons of cementitious composites with plain and functionalized nanotubes were made at the same concentrations, with the same methods of preparation and under the same conditions of testing. A focus on observations related to the mechanisms underlying the enhancement of mechanical strength and conductivity helps to clarify the benefits of using functionalized MWCNTs.

## 1. Introduction

Cementitious materials played a key role in the urbanization of modern economies, with utilization in the construction of commercial and residential buildings. Today, concrete remains an essential element in building construction and in the development of transportation infrastructure as an integral component to bridges, overpasses and pavement, such as in the interstate system of the United States. The importance of these materials has meant ongoing efforts to improve the properties of binder and concrete and to develop new uses with sustainability in mind. Many recent advances in the properties of cement paste, mortar and concrete have been through the incorporation of nanoparticles (NPs). Several reviews describe the benefits of composites with NPs of SiO_2_, TiO_2_, Al_2_O_3_ and other oxides which can increase strength under compression and tension [1,2,3,4,5,6].

Carbon nanotubes stand out as a subgroup of NPs because of their distinctive structure and remarkable features including high aspect ratio and electrical conductivity. Composed of an array of unsaturated hexagonal carbon rings, carbon nanotubes exist in zigzag, armchair and chiral forms (Figure 1). Multi-walled carbon nanotubes (MWCNTs) consist of a series of concentric, coaxial cylinders with diameters typically 20–50 nm, much larger than the 1 nm diameter of a single-walled carbon nanotube. These small diameters mean a high aspect ratio for lengths as high as 10 microns or more [7,8].

Due to lower cost and greater resistance to compression, MWCNTs are more widely used in composites than in single-wall forms. With tensile strengths on the order of 50–150 GPa [9,10], far greater than steel, carbon nanotubes offer the prospect of mitigating susceptibility to failure under tension, a well-known weakness of cementitious composites. Indeed, carbon nanotubes, with their high tensile strength, have shown the ability to reduce cracking and improve the flexural stress of mortars and cement pastes.

In addition to their physical strength, MWCNTs have high electrical and thermal conductivities of 2.2 × 10^4^ S/cm [11] and 3 × 10^3^ W/m K [12]. Their electrical conductivity supports features like electromagnetic shielding, dissipation of static charge and the piezocharacter that forms the basis for structural health monitoring. With high thermal conductivity, carbon nanotubes may help to reduce the degradation of structures at extreme temperatures, though the temperature dependence of MWCNTs thermal conductivity poses a challenge for this application [13]. These extraordinary properties make MWCNTs promising components for composites.

**Figure 1 nanomaterials-14-00080-f001:**
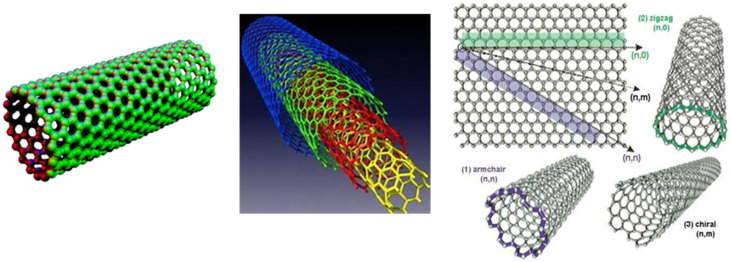
Multi-walled carbon nanotubes have been used more frequently than single-walled due to lower cost. Properties like electrical conductivity vary with the zig-zag, armchair and chiral chemical structures [14].

Interest in cementitious materials with MWCNTs has been high, with many articles appearing, and comprehensive reviews available [15,16,17]. Researchers have focused on the mechanical properties of compressive and flexural strength in cement paste and mortar composites with carbon nanotubes. However, findings in the literature have been inconsistent as to the extent of any benefit or even whether there is a benefit with functionalization. The addition of plain MWCNTs to cement paste [18,19,20,21,22] or mortar [23] has been shown to raise the compressive strength, while others have observed decreases or no change for cement paste and mortar composites [24,25,26,27,28]. Flexural strength with pristine MWCNTs at concentrations of 0.025–1.0 wt% of cement can increase as much as a factor of 2–3 [8] for cement paste, though increases are more typically 20–30% [18,20,29,30,31], and to a degree in mortar [23,32] as well. Yet, there are also reports with little or no improvement in flexural strength [33,34] for concentrations of 0.015–0.5 wt%.

Understanding the nature of carbon nanotubes led some researchers to propose functionalization of the nanotubes for better performance. The proposal considered the hydrophobicity of bare nanotubes, the interaction of MWCNTs with hydration products of cement, the aqueous processing conditions required and the mechanisms underlying the enhanced strength of cement composites. Chemical modification has shown promise, with the most common methods introducing hydroxyl and carboxylic acid groups to improve properties. Tests on composites with functionalized MWCNTs have yielded increases as high as 50% in the compressive strength for both cement paste [10,27,33,35] and mortar [22,36,37,38,39]. Similar levels of improvement for flexural strength are known for paste [10,33,40,41] and mortar [27,36,38,41]. While the modification of MWCNTs can certainly be rationalized, there are potential problems. The chemical reactions can damage the structure of the carbon nanotubes, weakening the properties that undergird the potential benefits. Some investigators found that functionalization yields comparable or worse performance of mechanical strength than bare MWCNTs [26,39,42,43]. Altogether, these reports create uncertainty about the merits of using functionalized carbon nanotubes.

It is clearly possible to improve mechanical strength with both nonfunctionalized and functionalized carbon nanotubes. There is, however, great variability in the results with pristine MWCNTs, and in the findings with functionalized MWCNTs as well, so reported values for compressive and flexural strength do not consistently and clearly favor functionalization. As such, the benefits of functionalization are not readily apparent. The failure in many investigations of functionalized nanotubes to include pristine nanotube composite samples for comparison has clouded the picture and not helped to resolve the question.

The purpose of this study is to examine the hypothesis that nanotube covalent functionalization improves mechanical and other properties of cementitious composites better than pristine nanotubes. The question is significant because the treatment of added carboxylic acid and alcohol substituents has been widely employed with the objective of improving dispersion and performance. For this review, an extensive search of the literature was conducted to find studies of cementitious composites incorporating functionalized carbon nanotubes. Of the many reports in the literature, articles were selected for inclusion and analysis based on a single, simple objective criterion of whether a pristine nanotube control was included. The approach yielded results for samples with pristine and functionalized carbon nanotubes from the same lab. These studies are more likely to have MWCNTs before and after functionalization of comparable structure and purity, consistent methods of dispersion, and similar techniques for sample preparation and testing. The scope is not limited to how functionalization affects mechanical strength. It includes the effect of functionalization on the mechanisms and properties underlying improved mechanical strength with MWCNTs with the same criterion applied in the selection of studies. Findings for properties of cementitious composites associated with strength (e.g., porosity and aggregation) are insightful. The results produce a clearer picture of the functionalization and mechanical strength of cementitious composites. As for electrical properties, fewer studies exist where direct comparisons have been conducted, but findings from direct comparisons of electrical conductivity are also presented. The addition of pristine or functionalized MWCNTs does reduce resistance or increase the conductivity of cement pastes [44,45,46,47,48,49] and mortars [50,51], though not always [52]. To affect this property, the MWCNTs must reach a minimum concentration for connectivity. The high aspect ratio of the nanotubes acts to lower the concentration of the percolation threshold over other fillers [53].

## 2. Functionalization of Multiwalled Carbon Nanotubes

The term functionalization has been used broadly to represent various methods to cause disaggregation of carbon nanotubes leading to their suspension in water with the objective of effective mixing to obtain distributed, better integrated nanotubes in the final composite. Functionalization changes the surface of the carbon nanotubes and helps to address the challenge that arises from the hydrophobic chemical nature of MWCNTs. Hydrophobicity impedes the distribution of the nanotubes in water, and ultimately in the composite, as the aqueous media is combined with cement and mixed. Not only are nanotubes not drawn into aqueous media, but their high surface energy with strong van der Waals attractive forces causes them to resist separation and to reaggregate during mixing. Distinct nanotubes in process water help ultimately to achieve separated nanotubes throughout the cementitious composite after processing. This is important because the disaggregated nanotubes, whether functionalized or not, contribute to mechanisms underlying strength improvement.

### 2.1. Methods of Functionalization

There is some ambiguity in what functionalization means to different researchers. Broadly applied, it includes the use of surfactants and polymers to improve the dispersion of MWCNTs in aqueous media to achieve better mechanical properties of pastes and mortar. Many, however, consider it limited to methods leading to the addition of covalently bonded polar substituents to the nanotubes. In this review, our use of functionalization focuses on nanotubes with covalently linked polar substituents. A common method to add alcohol and/or carboxylic acid groups employs a mixture of nitric and sulfuric acids. Other oxidizing conditions for functionalization have been treatment with KMnO_4_/H_2_SO_4_ [34], exposure to a low-temperature oxygen plasma [34] and reaction with ozone [49]. Singer et al. compare methods of oxidation and recommend a milder treatment with hydrogen peroxide [54]. Harsh conditions can cause degradation, so purification procedures after oxidation can be important. Removal of carboxylated carbonaceous fragment byproducts from the oxidation reaction has been reported to have a great effect on performance [25]. In some studies, the method of functionalization is not specified, only a supplier is indicated. MWCNTs purchased from companies may be superior in quality, with more time invested by the company in optimizing preparation and purification after functionalization.

### 2.2. Surfactants

The challenges to the dispersion of nanotubes in water can be successfully overcome with surfactants and some polymers [55,56,57]. The choice and amount of surfactant are important [58]. For example, the block copolymer surfactant Pluronic F127 outperformed the anionic sodium dodecylbenzene sulfonate to yield improved flexural and compressive strengths of mortar loaded with single-walled carbon nanotubes by 7% and 19%, respectively [59]. The optimal concentration of Pluronic F127 was determined from an absence of aggregation in optical micrographs. Konsta-Gdoutos et al. [29] reported that the optimum surfactant-to-CNTs mass ratio for a uniform dispersion is four, while Zou et al. [30] suggested a mass ratio of eight for achieving a good dispersion. Use of surfactants generally requires sonication, which means significant energy, extra time and possible damage to the nanotubes [58,60]. Li et al. [61] presented the mechanisms of surfactant-modified MWCNTs treated by ultrasound in an aqueous phase for the comparison of three different types of surfactants: anionic surfactant (sodium dodecyl sulfate, SDS), cationic surfactant (cetyltrimethylammonium bromide, CTAB) and nonionic surfactant (octylphenol polyoxyethylene ether, OP-10), as shown in Figure 2. The surfactants were employed to disperse the MWCNTs in aqueous solution with the assistance of ultrasonic waves. In comparison to anionic and nonionic surfactants (SDS and OP-10), the cationic surfactant (CTAB) yielded better dispersity of MWCNTs, because of greater electrostatic repulsion and higher steric resistance [61].

While it is possible to achieve favorable results for mechanical properties with surfactant [58,60], poor results are often obtained [25,28,62]. Surfactants may introduce voids, and weaken cementitious materials [63]. Siddique et al. found that the use of the surfactant sodium dodecyl sulfate in the preparation of an MWCNT mortar composite led to foaming and voids with less compressive strength [62]. Others have attributed a decrease in performance to surfactants carrying excess water into the cement matrix.

Treatment of MWCNTs to form -OH and -COOH groups would seem to offer advantages over surfactants. Functionalization of this type enhances the ability of water to break up aggregates with little use of sonication. Moreover, surfactants do not provide a covalent bond to the CNTs, so they are more susceptible to pullout under tension and less able to transfer forces within the cementitious composite [41]. Functionalization provides a basis for bonding as detailed below.

## 3. Mechanisms of Increased Mechanical Strength with Nanotubes and the Effects of Functionalization

If the hypothesis that functionalization of MWCNTs improves mechanical strength of composites better than pristine nanotubes is correct, then the difference should be evident in factors and mechanisms tied to mechanical behavior. In the sections below, properties known to affect compressive and flexural strength are examined. Results from head-to-head comparisons are emphasized for support or rejection of the hypothesis.

### 3.1. Dispersion and Functionalization

Dispersion is important for all NPs, but it is a particular challenge for carbon nanotubes. Functionalization alone will not achieve adequate dispersion. Mechanical energy input is needed with sonication to disaggregate the nanotubes in aqueous media and mix to form pastes and composites. In the absence of a means to overcome van der Waals forces, the addition of MWCNTs to cementitious materials can have a detrimental effect on mechanical properties. They must be spread throughout a cementitious composite to be effective, and that has proven to be difficult. For example, Malikov et al. observed a 27.8% drop in compressive strength with 0.01% pristine nanotubes that dispersed poorly in concrete [26].

By sonication of nonfunctionalized MWCNTs, Kumar et al. obtained a 15% increase in compressive strength and a 36% increase in tensile strength of a cement paste, that decreased at higher concentrations of nanotubes [20]. At higher concentrations, the drop in performance was attributed to poorer, nonuniform dispersion, which was evident in SEM imaging. Paste and mortar composites of both pristine and functionalized MWCNTs often exhibit decreasing compressive strength (Table 1A,B) above an intermediate weight percentage. Similar results have been reported for flexural strength (Table 2A,B). The reduction in mechanical strength occurring at higher concentrations is caused by reaggregation leading to agglomeration (Figure 3). In contrast, the effect of agglomeration on conductivity is not clear, due to the inconsistent trends of electrical conductivity (Table 3), though some data seem to show increasing conductivity over a wider concentration range of 0.05–2.0% [44,48,64]. Comparison is difficult, with concentrations in these studies based variously on the volume of cement paste, the weight percent of cement and the weight percent of the nanocomposite.

Functionalization improves dispersion. In comparative studies, functionalized MWCNTs by various methods displayed more effective dispersibility than untreated MWCNTs [10,25,34]. Superior dispersibility of modified MWCNTs in water has been demonstrated by UV-vis absorption, while SEM imaging of microstructure in the composite shows better dispersion stability in the cement matrix [34].

Whether functionalized or not, nanotube geometry and dimensions affect dispersion. Compared to short ones, longer, high aspect ratio nanotubes are more difficult to disperse [10,33,65], which can contribute to clumps of MWCNTs or agglomeration (Figure 3). Even so, long MWCNTs are more effective in improving strength, albeit short nanotubes at higher concentrations can achieve similar performance [8,33,66]. The diameter of the nanotubes is also a factor. Manzur et al. examined the compressive strength of cement composites for seven different sizes of nanotubes, with outer diameters (OD) ranging from less than 8 nm to greater than 50 nm and with lengths of 10–20 μm or 10–30 μm [39]. Both pristine and bare nanotube samples were tested. Composite compressive strength was greater for OD less than 20 nm with the smallest diameter functionalized sample yielding the highest value of 42 MPa for a 0.3% loading at day 28. The result was explained with smaller size nanotubes filling nanopore void space in the cement matrix more efficiently. At the same time, smaller diameters mean more surface area and a greater need to overcome surface energy requirements for dispersion.

**Figure 3 nanomaterials-14-00080-f003:**
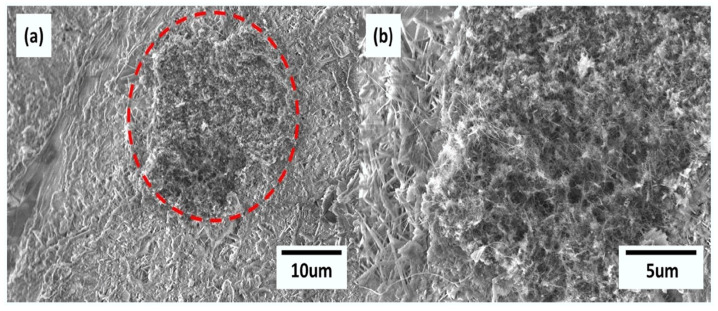
Agglomeration of bare MWCNTs in a cement paste composite [67]. MWCNTs reaggregate at higher concentrations during mixing to form agglomerates. The presence of these structures weakens the composite. An agglomerate in a composite is outlined in red (**a**) and shown at higher magnification (**b**).

Manzur et al. also concluded that the treatment of MWCNTs with a mixture of sulfuric and nitric acids introduced functional groups to reduce agglomeration and more uniformly disperse filler [39]. Similarly, Cui et al. found that functionalization led to better dispersion, significantly elevating the compressive and flexural strength of both long and short MWCNTs, relative to the improvement seen with pristine MWCNTs [10]. Hydroxyl-substituted nanotubes proved superior to carboxyl substituents in their study, though both produced excellent results.

**Table 1 nanomaterials-14-00080-t001:** (**A**) Comparison of compressive strength for cement paste with functionalized, pristine or no MWCNTs *. (**B**) Comparison of compressive strength for cement mortar with functionalized, pristine or no MWCNTs *.

(**A**)						
**Materials** **Cure Time**	**Compression** **Control**	**Compression** **Pristine**	**Compression** **Functionalized**	**Concentration Range ^†^**	**Functional Group or Method of Functionalization ^‡^**	**Ref.**
MWCNTs Cement Paste (silica fume)@14d	~45 MPa	~52 MPa	~75 MPa (no surf) ~72 MPa (with surf)	0.1%	Nitric/Sulfuric Acid	[35]
MWCNTs Cement Paste Cure time?	36 MPa	13 MPa	66 MPa 71 MPa 65 MPa 59 MPa	0.02% of paste 0.03% 0.05% 0.09%	Nitric and Sulfuric Acids	[25]
MWCNTs Cement Paste@28d	96.0 MPa	96.5 MPa 93.8 MPa	101.1 MPa 100.8 MPa	0.05% of paste 0.1%	-COOH	[63]
Long MWCNTs Cement Paste@28d	~95 MPa	~115 MPa ~110 MPa ~112 MPa	~125 MPa ~158 MPa ~120 MPa	0.1% 0.5% 0.8%	-OH	[10]
Long MWCNTs Cement Paste@28d	~95 MPa	~115 MPa ~110 MPa ~112 MPa	~130 MPa ~145 MPa ~145 MPa	0.1% 0.5% 0.8%	-COOH	[10]
Short MWCNTs Cement Paste@28d	~95 MPa	~100 MPa ~115 MPa ~75 MPa	~90 MPa ~155 MPa ~95 MPa	0.1% 0.5% 0.8%	-OH	[10]
Short MWCNTs Cement Paste@28d	~95 MPa	~100 MPa ~115 MPa ~75 MPa	~95 MPa ~135 MPa ~85 MPa	0.1% 0.5% 0.8%	-COOH	[10]
MWCNTs Cement Paste@28d	54.1 MPa	58.9 MPa 67.9 MPa 64.7 MPa 57.9 MPa 57.6 MPa	64.1 MPa 66.6 MPa 56.4 MPa 55.3 MPa 53.4 MPa	0.015% 0.05% 0.1% 0.25% 0.5%	-COOH	[33]
MWCNTs Cement Paste@28d	54.1 MPa	58.9 MPa 67.9 MPa 64.7 MPa 57.9 MPa 57.6 MPa	64.4 MPa 66.1 MPa 62.1 MPa 57.6 MPa 57.7 MPa	0.015% 0.05% 0.1% 0.25% 0.5%	-OH	[33]
MWCNTs Cement Paste Cure?	48.6 MPa	50.8 MPa	54.5 MPa	0.1%	Nitric/Sulfuric Acid	[68]
(**B**)						
**Materials ** **Cure Time**	**Compression** **Control**	**Compression** **Pristine**	**Compression** **Functionalized**	**Concentration Range ^†^**	**Functional Group or Method of Functionalization ^‡^**	**Ref.**
MWCNTs Cement Mortar@28d	~46 MPa	~51 MPa ~49 MPa ~49 MPa	~52 MPa ~54 MPa ~50 MPa	0.05% 0.1% 0.2%	-COOH	[34]
MWCNTs Cement Mortar@28d	~46 MPa	~51 MPa ~49 MPa ~49 MPa	~53 MPa ~54 MPa ~57 MPa	0.05% 0.1% 0.2%	Low temperature plasma	[34]
MWCNTs Cement Mortar@28d	~37 MPa	~38 MPa ~38 MPa ~39 MPa ~37 MPa	~40 MPa ~40 MPa ~41 MPa ~35 MPa	0.1% 0.2% 0.3% 0.5%	-COOH	[40]
MWCNTs Cement Mortar@28d	~37 MPa	~38 MPa ~38 MPa ~39 MPa ~37 MPa	~40 MPa ~40 MPa ~40 MPa ~36 MPa	0.1% 0.2% 0.3% 0.5%	-COOH	[40]
MWCNTs Cement Mortar@28d	~37 MPa	~38 MPa ~38 MPa ~39 MPa ~37 MPa	~40 MPa ~42 MPa ~42 MPa	0.1% 0.2% 0.3% 0.5%	-COOH	[40]
MWCNTs Cement Mortar@28d	~37 MPa	~38 MPa ~38 MPa ~39 MPa ~37 MPa	~40 MPa ~40 MPa ~36 MPa ~33 MPa	0.1% 0.2% 0.3% 0.5%	-COOH	[40]
MWCNTs Cement Mortar@28d	72.1 MPa	88.8 MPa	82.1 MPa	0.05%	-COOH	[69]
MWCNTs Cement Mortar@28d	72.1 MPa	88.8 MPa	85.3 MPa	0.05%	-OH	[69]
MWCNTs Cement Mortar@14d	~46 MPa	~50 MPa	~51 MPa ~50 MPa	0.15%	-COOH -OH	[22]
MWCNTs Mortar with 30% fumed sllica@14d	~59 MPa	~56 MPa	~71 MPa ~69 MPa	0.15%	-COOH -OH	[22]
MWCNTs Mortar with 30% fumed sllica@14d	~59 MPa	~56 MPa	~71 MPa ~69 MPa	0.15%	-COOH -OH	[22]

* The tilde symbol “~” indicates values estimated from graphical presentations of results. ^†^ Weight percent of cement. ^‡^ Functional group if specified or method of modification otherwise.

**Table 2 nanomaterials-14-00080-t002:** (**A**) Comparison of flexural strength for cement pastes with functionalized, pristine or no MWCNTs *. (**B**) Comparison of flexural strength for cement mortar with functionalized, pristine or no MWCNTs *.

(**A**)						
**Materials Cure Time**	**Flexural** **Control**	**Flexural ** **Pristine**	**Flexural ** **Functionalized**	**Concentration ** **Range ^†^**	**Functional Group or Method of Functionalization ^‡^**	**Ref.**
Long MWCNTs Cement Paste@28d	~8.5 MPa	~13 MPa ~11.5 MPa ~12.5 MPa	~9 MPa ~11 MPa ~11 MPa	0.1% 0.5% 0.8%	-OH	[10]
Long MWCNTs Cement Paste@28d	~8.5 MPa	~13 MPa ~11.5 MPa ~12.5 MPa	~13 MPa ~12 MPa ~13.5 MPa	0.1% 0.5% 0.8%	-COOH	[10]
Short MWCNTs Cement Paste@28d	~8.5 MPa	~9 MPa ~10.5 MPa ~9 MPa	~11 MPa ~15.5 MPa ~8 MPa	0.1% 0.5% 0.8%	-OH	[10]
Short MWCNTs Cement Paste@28d	~8.5 MPa	~9 MPa ~10.5 MPa ~9 MPa	~10.5 MPa ~13.5 MPa ~10 MPa	0.1% 0.5% 0.8%	-COOH	[10]
MWCNTs Cement Paste (silica fume)@14d	~4.3 MPa	~4.8 MPa (with surf)	~6.5 MPa (no surf) ~5.7 MPa (with surf)	0.1% of paste	Sulfuric and nitric acid	[35]
MWCNTs Cement Paste Cure?	1.5 MPa	1.6 MPa	4.7 MPa	0.1%	Sulfuric and nitric acid	[68]
MWCNTs Cement Paste@28d	7.7 MPa	9.0 MPa 10.3 MPa 9.4 MPa 8.8 MPa 8.3 MPa	8.7 MPa 9.3 MPa 8.4 MPa 7.1 MPa 6.4 MPa	0.015% 0.05% 0.1% 0.25% 0.5%	-COOH	[33]
MWCNTs Cement Paste@28d	7.7 MPa	9.0 MPa 10.3 MPa 9.4 MPa 8.8 MPa 8.3 MPa	8.7 MPa 9.3 MPa 8.8 MPa 8.1 MPa 6.7 MPa	0.015% 0.05% 0.1% 0.25% 0.5%	-OH	[33]
(**B**)						
**Materials** **Cure Time**	**Flexural** **Control**	**Flexural ** **Pristine**	**Flexural ** **Functionalized**	**Concentration ** **Range ^†^**	**Functional Group or Method of Functionalization ^‡^**	**Ref.**
MWCNTs Cement Mortar@28d	~7 MPa	~7.9 MPa ~8.2 MPa ~7.2 MPa	~8.5 MPa ~9.6 MPa ~9.0 MPa	0.05% 0.1% 0.2%	-COOH	[34]
MWCNTs Cement Mortar@28d	~7 MPa	~7.9 MPa ~8.2 MPa ~7.2 MPa	~8.9 MPa ~8.1 MPa ~8.3 MPa	0.05% 0.1% 0.2%	Low temperature plasma	[34]
MWCNTs Cement Mortar@28d	10.3 MPa	13.3 MPa	12.1 MPa	0.05%	-COOH	[69]
MWCNTs Cement Mortar@28d	10.3 MPa	13.3 MPa	11.6 MPa	0.05%	-OH	[69]
MWCNTs Mortar with 30% fumed sllica@14d	~4.4 MPa	5.9 MPa	~6.6 MPa ~6.4 MPa	0.15%	-COOH -OH	[22]

* The tilde symbol “~” indicates values estimated from graphical presentations of results. ^†^ Weight percent of cement. ^‡^ Functional group if specified or method of modification otherwise.

**Table 3 nanomaterials-14-00080-t003:** Comparison of electrical conductivities for cement pastes with functionalized, pristine or no MWCNTs *.

Materials Cure Time	Electrical Control	Electrical Pristine	Electrical Functionalized	Concentration Range ^†^	Functional Group or Method of Functionalization ^‡^	Ref.
MWCNTs Cement Paste@28d	~2.03 × 10^−7^ S/cm	~2.54 × 10^−7^ S/cm ~3.09 × 10^−7^ S/cm ~3.97 × 10^−5^ S/cm ~9.02 × 10^−4^ S/cm	~4.46 × 10^−7^ S/cm ~8.15 × 10^−7^ S/cm ~8.56 × 10^–4^ S/cm ~5.20 × 10^−3^ S/cm	0.05% 0.1% 0.3% 0.5%	-C=O from PVAc -NH from PIn (Admicellar polymerization)	[45]
MWCNTs Cement Paste@28d	~1.32 × 10^−7^ S/cm	~3.97 × 10^−7^ S/cm ~3.82 × 10^−5^ S/cm ~6.94 × 10^−4^ S/cm	~8.46 × 10^−7^ S/cm ~5.96 × 10^−4^ S/cm ~3.12 × 10^−3^ S/cm	0.1% 0.3% 0.5%	-C=O from PVAc -NH from PIn (Grafting polymerization)	[46]
Short MWCNTs Cement Paste@28d	~200 Ω·m	~155 Ω·m ~130 Ω·m ~179 Ω·m	~162 Ω·m ~117 Ω·m ~190 Ω·m	0.1% 0.5% 0.8%	-COOH	[48]
Short MWCNTs Cement Paste@28d	~200 Ω·m	~155 Ω·m ~130 Ω·m ~179 Ω·m	~151 Ω·m ~178 Ω·m ~159 Ω·m	0.1% 0.5% 0.8%	-OH	[48]
Long MWCNTs Cement Paste@28d	~200 Ω·m	~150 Ω·m ~145 Ω·m ~149 Ω·m	~140 Ω·m ~100 Ω·m ~90 Ω·m	0.1% 0.5% 0.8%	-COOH	[48]
Long MWCNTs Cement Paste@28d	~200 Ω·m	~150 Ω·m ~145 Ω·m ~149 Ω·m	~210 Ω·m ~120 Ω·m ~130 Ω·m	0.1% 0.5% 0.8%	-OH	[48]
MWCNTs Cement Paste@30d	-	~610 ΔR/Ω ~600 ΔR/Ω ~575 ΔR/Ω	~596 ΔR/Ω - -	0.1% 0.5% 2.0%	-COOH	[70]
MWCNTs Cement Paste@28d& 90d	17.16 Ω·m 401.07 Ω·m	15.13 Ω·m 291.03 Ω·m	12.91 Ω·m 207.47 Ω·m	1%	H_2_SO_4_-HNO_3_	[49]
MWCNTs Cement Paste@28d & 90d	17.16 Ω·m 401.07 Ω·m	15.13 Ω·m 291.03 Ω·m	14.14 Ω·m 126.96 Ω·m	1%	Ozone (O_3_)	[49]
MWCNTs Cement Paste@28d & 90d	17.16 Ω·m 401.07 Ω·m	15.13 Ω·m 291.03 Ω·m	14.73 Ω·m 50.35 Ω·m	1%	O_3_-NaOH	[49]
MWCNTs Cement Paste@1d	3.7 × 10^6^ Ω·m	3.6 × 10^6^ Ω·m	3.0 × 10^6^ Ω·m, 15 min 3.3 × 10^6^ Ω·m, 30 min 1.9 × 10^6^ Ω·m, 45 min 1.1 × 10^6^ Ω·m, 60 min	0.1%	-COOH, Oxidation time	[68]

* The tilde symbol “~” indicates values estimated from graphical presentations of results. ^†^ Weight percent of cement. ^‡^ Functional group if specified or method of modification otherwise.

Findings for aspect ratio were consistent with the work of Mazur et al. Long MWCNTs (10–30 µm, outer diameter less than 8 nm) generally outperformed short nanotubes (0.5–2 µm, outer diameter less than 8 nm) in mechanical strength (Table 1A and Table 2A). Ahmed et al. noted the strong influence of aspect ratio on dispersion and reinforcement and calculated theoretical nanotube spacing for ideal dispersion and uniform distribution in cement paste. This analysis showed that spacing decreased with aspect ratio as well as concentration [33]. While nanotubes with smaller aspect ratios are easier to disperse, the greater theoretical spacing means fewer are available at crack surfaces and there is less strength from MWCNT-matrix bonding. It was concluded that high aspect ratio MWCNTs are desirable if well dispersed [33].

Other methods, less common than covalent functionalization by oxidation, have been reported to enhance dispersion. Isopropanol can be used to separate nanotubes with sonication before suspending them in water [71]. A recent report describes the use of sodium hydroxide to disperse MWCNTs in composites [72]. Li et al. found synergism with the use of polyvinyl alcohol latex to help disperse hydroxy-modified MWCNTs [73]. These recent papers indicate ongoing interest in addressing dispersion.

In summary, functionalized MWCNTs enhance dispersity better than bare MWCNTs. This is important because improved mechanical properties with nanotubes depend on effective dispersion. Direct comparative studies by several groups have shown that MWCNTs with covalent functionalization by oxidation improved dispersion with concomitant superior strength. Consistency for long and short MWCNTs with theoretical analysis for the effect of aspect ratio helps to validate the results for dispersity.

### 3.2. Porosity and Functionalization

The void volume of a cementitious material is described by its porosity, a major determinant of compressive strength. An important property, porosity, as well as pore size distribution, can be determined by Mercury Intrusion Porosimetry (MIP) and by the digital analysis of Environmental SEM (ESEM) images [74]. In general, the addition of MWCNTs to composites causes a reduction in porosity [35,63,74,75]. This reduction in porosity with MWCNTs can be understood as their filling the small voids between hydration products of cement (Figure 4), as well as contributing to nucleation [10,18,29,40,76].

Mechanical strength is closely related to the pore structure of the composites [75]. Pores create sites for crack propagation and failure, so enhancing the effect of MWCNTs on porosity is a mechanism substantiating the benefits of functionalization. Kang et al. attributed the observed greater increase in compressive and tensile strength with acid-treated nanotubes to lower porosity compared to pristine nanotubes. They examined the porosity of cement paste composites containing either 0.1 wt% plain nanotubes or 0.1 wt% acid-treated nanotubes for porosity [35]. In a head-to-head comparison, the cement composite containing acid-treated, functionalized nanotubes was found to have lower porosity and greater strength than the one with plain nanotubes [35]. Hu measured the porosities of cement pastes with 0.1 wt% nanotubes and found that pristine MWCNTs led to a porosity of 24.84%, compared to 15.70% for 0.1 wt% functionalized MWCNTs [63]. The reduction in porosity corresponded with greater strength for functionalized MWCNTs [38,63].

Composite strength depends more on the total size distribution rather than just the total pore volume [63]; macropores with diameters greater than 50 nm affect strength more than smaller pores [43,63]. As such, it is notable that the addition of nanotubes causes a greater percentage reduction in the presence of larger pores than smaller pores. Li et al. used MIP to demonstrate that 0.5% acid-treated carbon nanotubes produced a Portland cement mortar with a total porosity 64% lower than the control mortar sample, yielding a value of 10.8% [38]. This is compared to the important macropore porosity at 1.47%, representing a reduction of 82%. 

Porosity and strength are very sensitive to the concentration of nanotubes, whether functionalized or not [34,35,63]. As the amount of MWCNTs modified with an oxygen plasma [34] increased for concentrations of 0, 0.05, 0.1 and 0.2%, the cumulative pore volume decline indicated a continuing decrease in micropores, corresponding to a continual increase in compressive strength. In other studies, with varying concentrations of MWCNTs, the compressive strength of the composites tends to exhibit a maximum (Table 1A,B), then decreases due to the formation of agglomerates (Figure 3). Bare hydrophobic MWCNTs may reaggregate at higher concentrations to form agglomerates, leading to defects, and acting as voids [17,34,65] to lessen the improvement in mechanical properties. The formation of agglomerates to create voids has the same effect as higher porosity.

A different mechanism for the reaggregation of functionalized MWCNTs must exist in place of the van der Waals forces. Li et al. proposed that high levels of Ca^2+^ and other multivalent cations may bind with carboxylic acids on functionalized MWCNTs to provide a chemical basis for agglomeration [38]. The work by Ahmed et al. illustrates the effect of agglomeration as observed by SEM [33]. They found optimal compression strength values of 66–67 MPa for cement paste modified with either pristine, hydroxyl functionalized or carboxyl functionalized MWCNTs at levels of 0.05% [33]. Smaller increases at higher concentrations for each were attributed to agglomerates with samples approaching the unmodified paste compressive strength of 54.1 MPa.

With its effect on porosity and agglomeration, the concentration of nanotubes has also been shown to affect the properties of concrete, with decreases in mechanical strength reported for concretes with functionalized [43,77] and bare MWCNTs [65]. In working with UHPC containing bare CNTs, Jung et al. found that the mechanisms for decreasing mechanical performance observed above the critical incorporation concentration (CIC) were CNT agglomeration and formation of air voids [65].

Functionalized MWCNTs have been shown to be superior to pristine MWCNTs in reducing porosity in cement pastes. Studies finding lowered porosity also observed improved compressive strength.

### 3.3. Fracture Resistance, Bridging and Interfacial Bonding with Nanotubes

Another important mechanism by which MWCNTs increase the strength of cementitious composites involves load transfer [10] by distributing stresses in an extensive network. Nanotubes can bridge micro-cracks (Figure 5a) and voids, as first shown by Makar and Beaudoin [8] and observed subsequently by others [8,20,31,77]. Smaller size MWCNTs allow for finer dispersion and the ability to stop crack propagation faster than reinforcement with larger fibers [31], though long, well-dispersed nanotubes are more likely to span cracks, according to a theoretical model [33]. The presence of MWCNTs acts to increase the fracture resistance of the composite [17].

The mechanism requires strong interfacial bonding for effective bridging, though failure may still occur if well-anchored nanotubes break (Figure 5b) [41]. Nonfunctionalized MWCNTs lack chemical bonds with the hydration products of the composite, so the interface is weak [22]. As a result, the pullout of nanotubes can happen, undermining the reinforcement of the composite (Figure 5c) [41]. Pullout and weak bridging compromise fracture resistance. Poor interfacial bonding can be addressed by treatment to create functional groups, commonly -OH and -COOH moieties [41]. Hu found that increases in fracture toughness for 0.1 wt% pristine MWCNTs were only 11.4% over plain cement, compared to 19.4% for MWCNTs-COOH [63]. Computer simulations support the benefit of the interactions afforded by functionalization. Theoretical investigations using molecular dynamics of CNTs have shown enhanced adhesion with tobermorite that grows with the number of acid functionalities to promote hydrogen bonding and ionic bonding with divalent calcium [78,79,80]. Acid-treated MWCNTs have covalently integrated carboxyl and hydroxyl groups that can react with the C-S-H and Ca(OH)_2_ hydration products of cement to produce strong bonds [38]. Using SEM and FTIR, Li et al. obtained evidence showing that reactions occur between the carboxylic acid and the cement matrix [38]. A greater increase in fracture resistance with functionalized MWCNTs results from the bonding of carboxylate and alcohol groups with C-S-H hydration products [22]. Ionic bonds can form with calcium and carboxylate or with hydrogen bonding (Figure 6) [81]. The mechanism of stronger bonding along with bridging and pull-out, evident in the microstructure, has been used to explain the greater impact resistance of UHPC with -OH and -COOH modified MWCNTs [82].

Covalent links to the nanotube structure, like those for carboxyl and hydroxyl moieties, seem to be a key element. In the absence of good interfacial bonding between the MWCNTs and the cement matrix, slippage and pullout occur, undermining force transmission. This is the issue with surfactant. Cwirzen et al. described how MWCNTs tend to pull out under tension due to slippage with nanotubes lacking the covalently linked polar groups that come with functionalization [83]. The group noted, as an example, that while noncovalently linked polyacrylic acid could help with dispersion, it still had weak bond strength and was subject to pullout [83]. In the absence of polar groups on the nanotubes, they concluded in their study that bare MWCNTs did not increase the compressive or bending strength of cement paste.

Many factors can influence the interface bonding contributing to pullout. For example, increasing the site density of carboxylate groups on the MWCNTs can mean a greater number of bonds to the matrix to avoid slippage. Processing conditions resulting in insufficient hydration and poor wetting can weaken bonding, while other species can interfere with bonding. Carboxylate will not compete effectively against residual oxide (O^2−^) for bonding to Ca^2+^. Bonding can apparently be blocked, as Nasibulina et al. found that using surfactants in combination with functionalized nanotubes resulted in compromised compressive strength. They hypothesized that the surfactant blocked the interaction of the functional group with cement [25].

### 3.4. Nucleation and Hydration

According to some researchers, the high surface area of pristine MWCNTs creates more sites for nucleation and accelerates the hydration process [21,32,69,77,84,85]. Enhanced nucleation and hydration lead to the formation of more portlandite of higher crystallinity [33] and integration of MWCNTs with calcium silicate hydrate (C-S-H) (Figure 6). Moreover, the promotion of the hydration reactions to form C-S-H in small interstitial spaces leads to densification and a shift in pore size distribution that contributes to greater strength [10,21,35]. This densification occurs with both pristine [21,32,74] and functionalized MWCNTs [35], but, in a direct comparison, Kang et al. found the hydration products to be denser with the acid-treated nanotubes [35]. That result is consistent with reported strength enhancement due to reduced crack formation with accelerated hydration [84].

Others observing a decrease in mechanical strength with functionalized MWCNTs have attributed the loss to lower hydration and a shift in reaction products. Cui et al. found that the degree of hydration decreased with functionalized MWCNTs, though to a lesser degree than with pristine nanotubes [10]. This can be explained by functionalized nanotubes being so hydrophilic that they absorb water to impede hydration.

The presence of functionalized MWCNTs can alter the reaction products of hydration. Ahmed et al. suggested more formation of ettringite [33] with functionalized MWCNTs, due to the presence of sulfate from acid treatment [34], thereby causing a less dense structure of the hydration products and poorer mechanical properties. Similarly, Musso et al. explained a reduction in the compressive and flexural strength of a cement paste upon the addition of carboxyl MWCNTs with poorer hydration and less formation of tobermorite [42]. Pristine CNTs, on the other hand, have been reported not to accelerate or change hydration reactions [86].

## 4. Electrical and Thermal Conductivity of Cementitious Composites with Functionalized and Pristine MWCNTs

Fewer studies of composites were found for electrical conductivity, and none were found for thermal conductivity, using the criterion of a pristine MWCNT control sample. It is known that the thermal conductivity of cementitious composites improves with the addition of nanotubes [12,81,87] for both pristine and functionalized MWCNTs.

Dispersion of MWCNTs during the hydration reaction of cement is important to developing a network structure inside the cement matrix to enhance the electrical and thermal conductivity [64]. Better electrical conductivity is found with higher MWCNT loading (0.05 to 0.8 wt% of cement) in accordance with a denser network of MWCNTs inside the cement (Figure 7) [45,48,64,70]. When mixing the MWCNT suspensions with the cement, one portion of the dispersed MWCNTs will be incorporated into the hydrating cementitious matrix, particularly during the formation of C-S-H and ettringite. This helps the cured composite to attain the critical distance of 2–3 nm for electron tunnelling between individual MWCNTs in a denser MWCNT network [64]. Ruan et al. found no clear difference in the resistance of cement paste between short MWCNTs with either short hydroxyl or short carboxyl functionalized nanotubes [48]. However, cement pastes with functionalized long MWCNTs did outperform the composite with pristine carbon nanotubes (Table 3). The lower resistivity of the long MWCNTs compared to the short MWCNTs is consistent with smaller theoretical spacing for a higher aspect ratio [33]. Onthong et al. modified the surface of MWCNT by coating it with a conducting polymer, polyindole and polyvinyl acetate using concurrent admicellar polymerization [45]. The coated MWCNTs obtained provided good water dispersion and yielded a cement paste (0.3 wt% MWCNTs) with an electrical conductivity of 8.56 × 10^−4^ S/cm, more than 20 times greater than the paste with bare MWCNTs. With the exception of the inconclusive results for the short MWCNTs, the findings overall in Table 3 indicate increased conductivity or reduced resistivity with covalent functionalization.

Thermal conductivity of cementitious composites improves on the addition of nanotubes [12,81,87]. To the authors’ knowledge, there are no studies where the thermal conductivity of cementitious composites was compared for bare and functionalized MWCNTs. The thermal conductivity of grouting material rose steadily with concentration, from 0.39 W/m-K with no nanotubes to 0.57 W/m-K at the concentration tested of 2 wt% [81]. Batiston et al. investigated thermal conductivity of cement pastes with pristine MWCNTs as a function of concentration and aspect ratio. At 0.05%, experimental values increased from 0.73 W/m-K for the control to an optimum of 0.84 W/m-K at an aspect ratio of 250. Values were lower for a loading of 0.10%.

## 5. Conclusions

Addressing the value of covalent functionalization is significant because it has been a common technique to enhance performance of cementitious composites. The question of the value of functionalization is understandable, with complex processes for cementitious composites with many options for components, composition and methods of preparation. Methods of dispersion vary with whether to use surfactant, type and concentration of surfactant and power and duration of sonication. Aside from the question of functionalization or not, carbon nanotubes come in different sizes and purity which can have a significant effect. With so many factors and seemingly conflicting reports, uncertainty related to functionalization is not surprising. The selection of studies for analysis based on the inclusion of a pristine MWCNT control reduced ambiguity and led to the following observations:(1)Both covalently functionalized and bare nanotubes improve the compressive and flexural strength of cement paste and mortar.(2)Covalent functionalization by oxidation of MWCNTs leads to a greater improvement in mechanical strength. Analyzing the tabulated values supports the superiority of functionalized MWCNTs. The average increase in compressive strength among different research groups for cement paste (Table 1A) is about 14% for pristine and 34% for functionalized. For mortar (Table 1B), the numbers are 11% and 16%. For flexural strength of the paste, pristine MWCNTs yield an average 23% increase compared to 43% for functionalized (Table 2A); for mortar (Table 2B), the results are 28% and 32%.(3)Greater reduction in porosity, greater increase in dispersity and greater fracture resistance occur with functionalized MWCNTs. These changes align with the improved mechanical properties in cementitious composites.(4)Data for cement pastes show that composites with functionalized MWCNTs have higher electrical conductivity than those with pristine MWCNTs. The aspect ratio appears to be a critical factor for conductivity, though additional work is needed. The average reduction of tabulated values of resistivity for cement paste is about 35% for pristine MWCNTs and 50% for functionalized.

Overall, these suggest that functionalized nanotubes are superior to pristine nanotubes. Trends in properties that affect mechanical strength reinforce the direct measurements of improved compressive and flexural strength. The electrical conductivity of cementitious composites is also enhanced with covalent functionalization. Thus, we conclude that a direct comparison of mechanical strength, as well as electrical conductivity, makes a strong case for using functionalized nanotubes where performance alone is considered.

## Figures and Tables

**Figure 2 nanomaterials-14-00080-f002:**
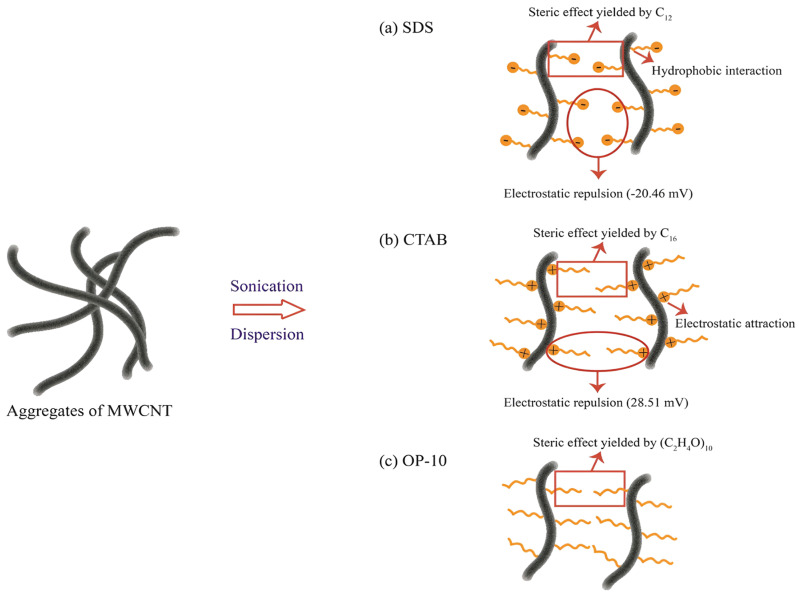
Interactions between MWCNT and surfactant with ultrasonic treatment [61]. Adsorbed surfactant on the surface of nanotubes acts to overcome van der Waals forces and facilitates disaggregation with ultrasound. Anionic surfactants like sodium dodecyl sulfate (SDS), and cationic surfactants like cetyltrimethylammonium bromide (CTAB), work by repulsion of like charges and steric effects. The nonionic octylphenol othoxylate (OP-10) functions by steric interactions only.

**Figure 4 nanomaterials-14-00080-f004:**
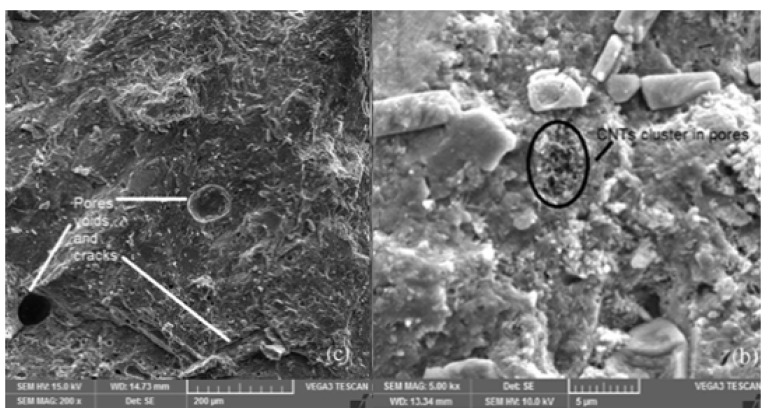
SEM images of mortar at 200× show pores (**c**), and at 5000× magnification (**b**) show a pore filled with plain MWCNTs [22]. MWCNTs can fill pores and reduce porosity to increase the compressive strength of cement pastes and mortars.

**Figure 5 nanomaterials-14-00080-f005:**
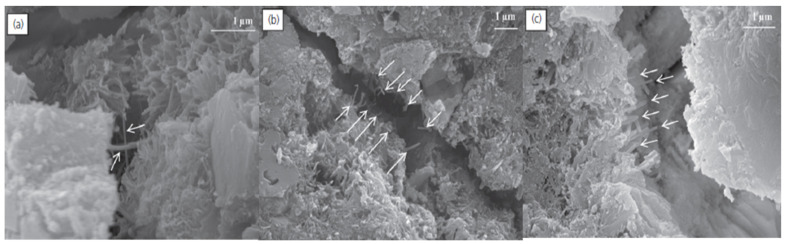
SEM images showing (**a**) carboxyl functionalized nanotubes bridging a crack, (**b**) broken nanotubes and (**c**) pullout of pristine MWCNTs [33]. Nanotubes well anchored to the hydration products can inhibit crack propagation but may fail due to weak bonding or breakage.

**Figure 6 nanomaterials-14-00080-f006:**
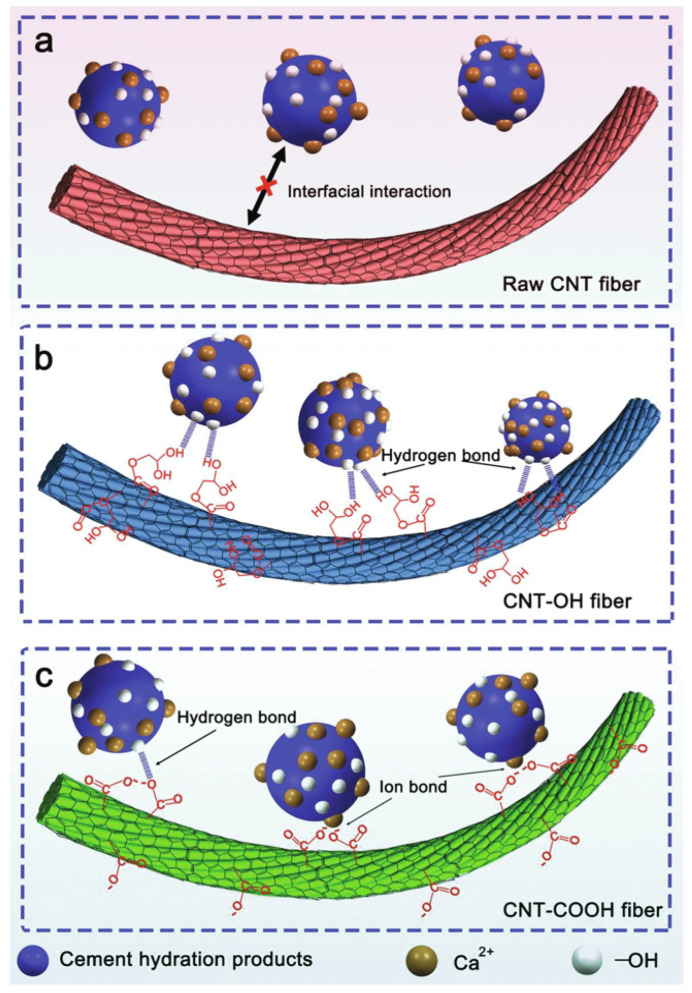
Comparison of interfacial interaction mechanism between cement hydration products with (**a**) bare MWCNT, (**b**) MWCNT-OH and (**c**) MWCNT-COOH [81]. Alcohol functional groups can hydrogen bond to C-S-H while carboxylic acid functionality can form both hydrogen and ionic bonds.

**Figure 7 nanomaterials-14-00080-f007:**
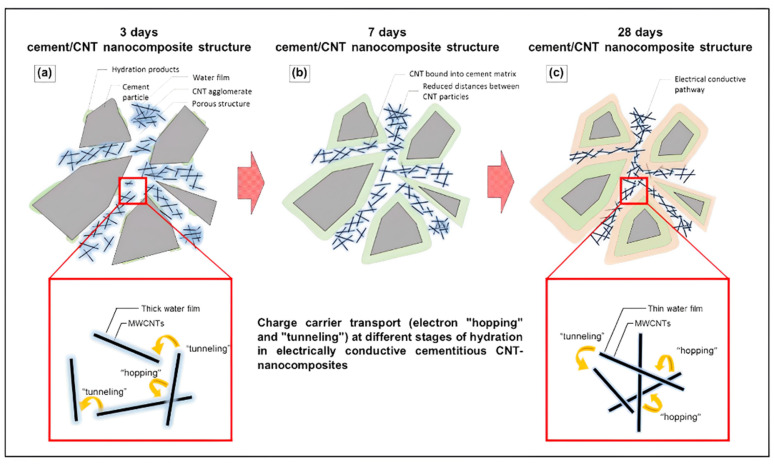
Scheme of the MWCNT interconnected network within a cementitious matrix and the occurring charge-carrier mechanisms (hopping–tunnelling) as a function of time [64]. Liebscher et al. describe how increasing density of carbon nanotubes in a composite changes their proximity and alters the contribution of hopping and tunneling to charge transport.

## Data Availability

Data are contained within the article.

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
