# Peer review of "Mechanical Strength and Conductivity of Cementitious Composites with Multiwalled Carbon Nanotubes: To Functionalize or Not?"

_nanomaterials, 2023, doi:10.3390/nano14010080_

Round 1
Reviewer 1 Report (Previous Reviewer 1)
Comments and Suggestions for Authors
The mini-review about functionalization of Carbon nanotubes could be interesting for future development of the technology and for other research works. The work could be published after major review. Some gaps of relevance, identified in the attached file, should be solved.
The reviewer recommends to clearly indicate in the conclusion the intent and the significance of the paper related to possible applications.
The reviewer recommends to correct the typos throughout the paper, with particular attention to the type of character, superscripts, subscripts and spaces. Moreover, the reviewer suggests to avoid the use of symbols in the text (as for example <, >)
The reviewer suggests to check the copyright of the figures and ask for permission if needed.
Further suggestions in the revised manuscript.
The mini-review about functionalization of Carbon nanotubes could be interesting for future development of the technology and for other research works. The work could be published after major review. Some gaps of relevance, identified in the attached file, should be solved.
The reviewer recommends to clearly indicate in the conclusion the intent and the significance of the paper related to possible applications.
The reviewer suggests to check the copyright of the figures and ask for permission if needed.
Further suggestions in the revised manuscript.

The reviewer recommends to correct the typos throughout the paper, with particular attention to the type of character, superscripts, subscripts and spaces. Moreover, the reviewer suggests avoiding the use of symbols in sentences (for example <, >)
Author Response
Please see the attachment

Reviewer 2 Report (Previous Reviewer 3)
Comments and Suggestions for Authors
The key issues of introducing MWCNTs into composite materials, for example, functionalizing, mechanics, durability, resilience, resistance to van der Waals force, and evenly dispersion are not well considered. The author did not review articles with unfavorable factors, the results resented in this research are too idealistic.
Comments on the Quality of English LanguageN.A.
Author Response
Please see the attachment

Reviewer 3 Report (New Reviewer)
Comments and Suggestions for Authors
1 {Summary}
This paper is based on multiple literature reviews and compares the mechanisms of increased mechanical strength and conductivity of functionalized nanotubes and bare nanotubes when used in cementitious composites. The main contribution of this paper is to directly compare the dispersion, porosity, fracture resistance and hydration of composites with functionalized and bare nanotubes. It starts from the main characteristics that affect mechanical properties. Visually demonstrate the differences and connections between the two.
2{Strengths and Weaknesses}
Strengths
(1) The authors consult a lot of literature and provide sufficient background knowledge.
(2) This paper's motivation and main contributions are clearly explained.
Weaknesses
(1) Some images are not clear and do not provide explanations in detail.
(2) The table features and content are not consistent.
(3) Limited contribution and novelty.
3 {Questions for the Authors}
(1) The title of paper emphasizes mechanical strength and conductivity. Based on the title, it can be seen that they are two entry points for discussing whether to functionalize. Therefore, they should be explained separately. Why is the part of conductivity (3.5) placed in the chapter of mechanical strength mechanisms (3) instead of explaining them separately.
(2) The comparison of data in this paper seems not convincing enough. Note that the conclusion mentions “Aside from the question of functionalization or not, carbon nanotubes come in different sizes and purity which can have a significant effect.” In all the tables, the authors should clarify whether the specifications of carbon nanotubes used in these experiments from different literature are consistent, and whether the significant impact of these part need to be considered.
(3) The paper mentions the contribution of carbon nanotubes to tensile strength in multiple places. Such as “Indeed, carbon nanotubes with their high tensile strength have shown the ability to reduce cracking and to improve the flexural stress of mortars and cement pastes” in introduction and “… a 36% increase in tensile strength of cement paste…”in 3.1. These indicate that tensile strength may also be the main factor in mechanical strength. This reviewer wants to know whether it is necessary to add actual experimental data on tensile strength from the literature. If not necessary, what are the reasons.
(4) Surfactant is also one of the methods for functionalizing carbon nanotubes. Why the author separates "Surfactants" from "Methods of Functionalization".
(5) Mechanical strength and conductivity are the two main research objectives of the paper. The conclusion summarizes and compares the data in Tables 1 and 2 about mechanical strength, but does not see the analysis and summary of the data in Table 3 about conductivity.
Comments on the Quality of English LanguageModerate editing of English language required
Round 2
Reviewer 1 Report (Previous Reviewer 1)
Comments and Suggestions for Authors
The authors have appropriately addressed almost all the reviewer's suggestions, and the paper has reached a sufficient level for publication.
Comments on the Quality of English LanguagePlease check and correct all typos (e.g. spaces, superscripts/subscripts)
Author Response
Thank you again for your comments. They have been helpful in improving the paper.
Reviewer 2 Report (Previous Reviewer 3)
Comments and Suggestions for Authors
reject.
Comments on the Quality of English LanguageN.A.
Author Response
Response to Reviewer 2:
You made initial comments on the scope and depth of our paper. There are topics on which we agree. Our manuscript does have whole sections devoted to functionalization and dispersion. We discuss the significance of overcoming the van der Waals force and we examine mechanics through the properties of compressive strength and flexural strength. To include other topics, however merit worthy, presumes publications exist that meet our criterion for inclusion.
The authors did examine publications with unfavorable findings. Our data is not perfect as some entries in the tables show. Some other articles with negative results were inconclusive as they lacked a control sample (lines 86-88, references 26, 39, 42 and 43) for comparison. Their data was not used. Similarly, other articles with favorable results were inconclusive for the same reason and not included.
Both the data and the process for identifying valid data have been described so the reader can decide for himself the merits of our case.
This manuscript is a resubmission of an earlier submission. The following is a list of the peer review reports and author responses from that submission.
Round 1
Reviewer 1 Report
Comments and Suggestions for Authors
The paper is about the state-of-the-art of functionalized Carbon nanotubes. Although the topic could be interesting, for further researches and investigations, the paper presents several drawbacks. There is a lack of scientific interpretation of the presented results, representing a key feature for the significance of reviews. The organization of the sections and of the text should be improved, in order to better identify the main aspects and the logical subsequence of the review procedure.
The reviewer recommends to reorganize the paper clearly indicating the different types of functionalization, and specific applications. As a matter of fact, some parts seems as a list of results of a few works, without a defined logical and critical sequence and evaluation. Even if the paper is defined as a “mini” review, it should be significant for a specific sector or intent.
Some comments are too generic. The authors should justify and prove the significance of the research.
The reviewer recommends to correct the typos throughout the paper, with particular attention to spaces, type of character and superscripts/subscripts.
The reviewer recommends to check the paper, modify the phrases that are not clear, and add comments if needed.
The reviewer suggests to check the copyright of the figures and ask for permission if needed.
In my opinion, the paper cannot be published as it is, and needs a strong reorganization and improvement.
Further suggestions in the revised manuscript.
The reviewer suggests to resubmit after modification.

There are lots of typos throughout the paper. Some parts are not clear. Sometimes the language could be more "scientific".
Reviewer 2 Report
Comments and Suggestions for Authors
It is a valuable work to review the role of incorporation of carbon nanotubes into cementitious composites on compressive and flexural strength as well as electrical and thermal conductivity. However, more exact data should be provided. For example:
(1)In looking at 7 different sizes of MWCNTs, Manzur et al. observed higher compressive strengths with smaller diameter nanotubes[38]. It had better to present the detailed sizes of MWCNTs, compressive strengths etc.
(2)Better electrical conductivity is found with higher MWCNT loading in accordance with a denser network of MWCNTs. Point out the MWCNT loading.
(3) Thermal conductivity of cementitious composites improves on addition of nano-tubes[12, 65, 75]. Increases have been reported for both pristine and functionalized MWCNTs. Add detailed data comparison.
Some corrections are needed.
Reviewer 3 Report
Comments and Suggestions for Authors
Review comments:
The article discusses the effect of MWCNTs functionalized on the mechanical strength and conductivity of cementitious composites,which is interesting and useful for researchers in related fields. The reviewer suggests that his paper can be published by considering the following issues in revision.
(1) Author(s) should make clear the purpose of the study and the specific questions to be explored in the introduction section, rather than simply stating and comparing the conclusions of the literature.
(2) Author(s) should add instructions on how to make cementitious composites, how to add MWCNTs, and how to perform mechanical strength and conductivity tests to make the research more practical.
(3) The conclusion part should present a more specific summary of the conductivity study and provide a further view on the research application of MWCNTs in cementitious composites.
(4) Tables should follow the textual description. e.g., Table 1A, Table 1B, Table 2A, Table 2B, Table 3A, and Table 3B should be placed in Part 2.1.
(5) Related related publications on micro and nano tubular structures are suggested to be reviewed in Introduction, for example, DOI: 10.1016/j.cma.2012.02.023
(6) Author(s) should revise the entire manuscript accordingly to bring the expression more in line with academic norms.
Comments on the Quality of English LanguageEnglish should be double checked.